# Dynamical analysis of rumor propagation model considering media refutation and individual refutation*

1st Wenqi Pan
*College of Marine Electrical Engineering*
*Dalian Maritime University*
Dalian, China
panwenqi07@163.com

2nd Li-Ying Hao*
*College of Marine Electrical Engineering*
*Dalian Maritime University*
Dalian, China
haoliying_0305@163.com

*Abstract*—The factor of refutation significantly impacts the spread of rumors. Common methods of refuting rumors include media intervention and individual efforts. While many scholars have explored the effects of these factors separately, few studies have comprehensively examined both simultaneously. This model integrates the influence of both media and individual refutation on the rumor propagation process. We propose a novel two-tier network model for rumor spread. We demonstrated the existence and stability of equilibrium points within the model. Theoretical analysis demonstrates that authoritative media refutations exert a broader and more substantial influence on rumor dissemination compared to individual refutations.

*Index Terms*—rumor propagation, rumor refuting medias, rumor refuters, stability

## I. INTRODUCTION

Rumor refers to the speech fabricated without corresponding factual basis and with a certain purpose and promoted its dissemination by some means. With the exponential growth of technology and the widespread adoption of internet-based social networks, misinformation and harmful rumors have the potential to swiftly propagate across online platforms, posing significant threats to social cohesion, stability, as well as disrupting people's daily lives and productive activities. For example, the panic of buying salt caused by the Fukushima Daiichi Nuclear Disaster [1], there was a rumor that SHL-C could prevent COVID-19, which caused great harm to the public's psychology and body, and seriously disturbed the normal order of the society.

The propagate of rumors has attracted the attention and research of many scholars. Some scholars compared the disseminate of rumors with the propagate of infectious diseases in humans, and applied the infectious disease model to the disseminate of rumors [2]–[5]. Considering the influence of different propagation mechanisms on rumor propagation, many scholars have studied cross propagation mechanism [7] and education mechanism [6]. Komi [8] established rumor propagation model based on population education and forgetting

This work was funded by the National Natural Science Foundation of China (51939001, 52171292), Dalian Outstanding Young Talents Program (2022RJ05).

mechanism, and found that educated ignorant people are less likely to be transformed into disseminators and more likely to be transformed into suppressors than uneducated ignorant people.

At the same time, many scholars considered the influence of different function methods [9]–[11] in the research process. Zhu et al. [14] proposed a rumor propagation model in homogeneous and heterogeneous networks, and comprehensively studied the influence of forced silence function, time delay and network topology on rumor propagation in social networks. The influence of time delay on the propagation process has also been studied by many scholars [15]–[18] on rumor propagation process in existing research. Cheng et al. [21] established an improved $XY - ISR$ rumor propagation model on the basis of interactive system, comprehensively discussed the influence of different delays on rumor propagation, further proposed control strategies such as deleting posts, popular science education and immunotherapy.

With the complexity of the network environment, some scholars have comprehensively considered the influence of various factors on rumor propagation on the complex network [22]–[24]. Considering the reaction of the ignorant when hearing the rumor for the first time, Huo et al. [25] divided the individuals in the network into four groups: the ignorant, the trustworthy, the spreader and the uninterested, and proposed $SIbInIu$ rumor propagation model in the complex network. The theoretical analysis and simulation results show that the loss rate and suppression rate have a negative impact on the final rumor spread scale.

In the existing literature, it is not common to comprehensively consider the impact of media refutation and individual refutation on the two-tier network rumor propagation model. Based on the actual assumptions, we believe that the rumor refutation effect of comprehensive consideration of the two is better than that of single consideration. This paper mainly make a dynamic analysis on the rumor propagation considering the rumor refutation effect of these two factors.

The rest of this paper is distributed as follows. We propose a two-tier network rumor propagation model in section 1.

Section 2 describes a two-tier network rumor propagation model considering both rumor refuting media and rumor refuter groups. In section 3, we discuss the existence and stability conditions of the equilibrium points. Finally, the feasibility of the results presented in this paper was confirmed through numerical simulations.

## II. TWO-TIER NETWORK RUMOR PROPAGATION MODEL

In the two-tier rumor propagation model constructed in this paper, the media network model is composed of networks with $M$ media websites, and the personal friendship network model is composed of networks with $N$ personal friendship websites.

In the network layer of media websites, media can be divided into three states: vulnerable media without rumor information (represented by $X$), affected media with rumor information (represented by $Y$) and rumor refuting media with rumor refuting information (represented by $Z$). When communicators visit the vulnerable media, they will release or leave rumors on the media network, so that the vulnerable media will be affected and become the affected media. When the rumor refuters visit the affected media, they will release or leave rumor refutation information on the media network to make the affected media become rumor refutation media.

In the personal network layer, individuals are categorized into four distinct groups: those who have never heard of the rumors (denoted by $S$), those who actively spread rumors (denoted by $I$), those who do not believe in the rumors but disseminate refutation information (denoted by $D$), and those who neither believe in nor propagate any information (denoted by $R$). In the process of network node interaction, after visiting the vulnerable media, the disseminator spreads rumor information on the media website, so that the vulnerable media is infected and evolved into the affected media. When an ignorant person visits the affected media, affected by the rumor information, the ignorant person becomes a disseminator with a certain probability. Thus, rumors can be spread not only between people, but also between individuals and online media. The basic assumptions of this paper are as follows:

Hypothesis 1: In the media network layer, considering that the media website has a certain registration rate and cancellation rate, the number of vulnerable media entering the communication system per unit time is $\Lambda_1$. Moreover, there will be benign competition among the media. The three types of media websites $X$, $Y$ and $Z$ may move out of the communication system with a certain probability $\mu_1$. When communicators visit vulnerable media and publish their own views and comments, the rate of conversion to affected media is $\lambda$. When the rumor refuter visits the affected media and publishes rumor refutation information on it, the affected media will change into rumor refutation media with a certain probability $\eta$.

Hypothesis 2: In the personal interpersonal network layer, assume that the rate at which individuals who are unaware of rumors enter the communication system is $\Lambda_2$. Those who question the rumor but neither spread rumor information nor disseminate refutation will transition to an immune state at a rate of $\xi_2$. Individuals who initially spread rumors but later find the information untrue may become rumor disclaimers with probability $\delta$. If these communicators lose interest in rumors and cease both rumor propagation and refutation, they will transition to an immune state with probability $\theta$. Rumor disclaimers affected by the environment or who lose interest in refutation will also become immune with probability $\phi$. Additionally, individual groups may exit the rumor spreading network due to migration at a rate $\mu_2$.

Hypothesis 3: In the interaction of offline individuals, the ignorant will become the disseminator at a certain rate $\alpha$ after contacting the disseminator. If ignorant person believe and propagate rumors after visiting the affected media, they will become disseminators at a certain rate $\beta$. It is assumed that after the unknown person contacts the rumor information (including contact with people and knowing the rumor information from the media), they realize that the rumor information is untrue due to them own experience or discrimination ability. If an individual who is initially unaware of the rumors chooses to disseminate rumor refutation information, they will transition to the status of a rumor refuter at a rate of $\xi_1$.

Based on the above analysis, the rumor propagation process of $XYZ - SIDR$ model established in this paper is shown in Fig. 1.

The meanings of symbols in Fig. 1 are shown in the following table. I.

TABLE I
DESCRIPTION OF PARAMETERS IN THE MODEL

| Parameter | Description |
|---|---|
| $\Lambda_1$ | The number of susceptible media entering the communication system per unit time. |
| $\Lambda_2$ | The number of ignorant individuals entering the communication system per unit time. |
| $\lambda$ | The contact rate of susceptible medias with spreaders. |
| $\eta$ | The probability of affected media becoming rumor refuting media. |
| $\alpha$ | Rumor propagation rate of offline personal interaction. |
| $\beta$ | Rumor propagation rate under two-tier network interaction. |
| $\delta$ | The probability of propagating individuals becoming rumor refuting individuals. |
| $\theta$ | The probability of propagating individuals becoming immune individuals. |
| $\xi_1$ | The rate of ignorant individuals becoming rumor refuting individuals. |
| $\xi_2$ | The rate of ignorant individuals becoming immune individuals. |
| $\phi$ | The probability of rumor refuting individuals becoming immune individuals. |
| $\mu_1$ | The rate at which medias in the network move out of the propagation system. |
| $\mu_2$ | Migration rate of individuals in personal friendship network layer. |

Based on the above analysis, we participated in the construc-

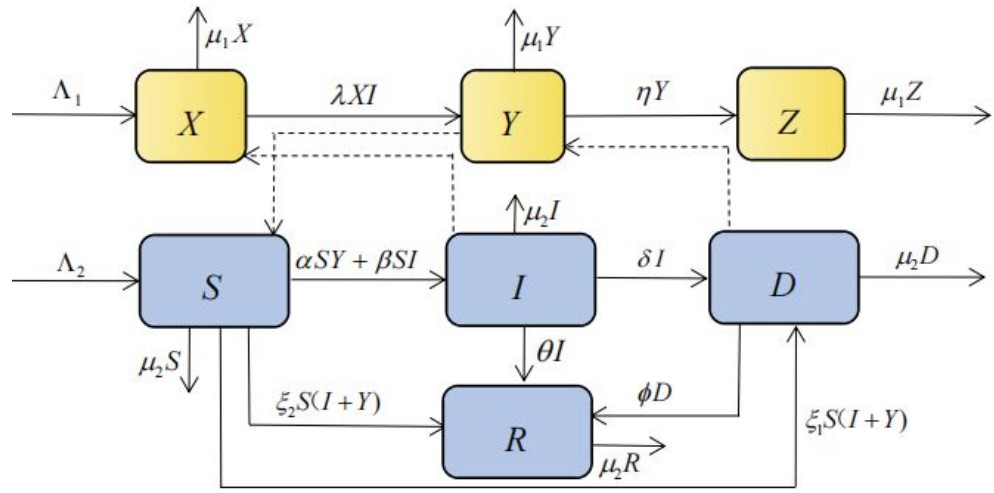

Fig 1. Schematic representation of the $XYZ - SIDR$ rumor spreading model

tion of an $XYZ - SIDR$ rumor propagation model. Then,

$$\begin{cases} X' = \Lambda_1 - \lambda XI - \mu_1 X, \\ Y' = \lambda XI - \eta Y - \mu_1 Y, \\ Z' = \eta Y - \mu_1 Z, \\ S' = \Lambda_2 - \alpha SY - \beta SI - (\xi_1 + \xi_2)(I + Y)S - \mu_2 S, \\ I' = \alpha SY + \beta SI - (\theta + \delta)I - \mu_2 I, \\ D' = \xi_1 S(I + Y) + \delta I - \phi D - \mu_2 D, \\ R' = \xi_2 S(I + Y) + \theta I + \phi D - \mu_2 R. \end{cases}$$
(1)

Since the model represents the process of rumor propagation, the parameters involved are non negative, and the initial conditions are met:

$$X(0) = X_0 \geq 0, Y(0) = Y_0 \geq 0, Z(0) = Z_0 \geq 0,$$
$$S(0) = S_0 \geq 0, I(0) = I_0 \geq 0, D(0) = D_0 \geq 0, \quad (2)$$
$$R(0) = R_0 \geq 0.$$

### III. MODEL ANALYSIS AND CALCULATION

#### A. The basic reproduction number $R_0$

For system (1), the basic regeneration number $R_0$ is calculated as follows:

Let $\mathscr{X} = (I, Y, R, D, S, X, Z)^T$, equation (1) can be written as $\frac{d\mathscr{X}}{dt} = \mathscr{F}(\mathscr{X}) - \mathscr{V}(\mathscr{X})$.

$$\mathscr{F}(\mathscr{X}) = \begin{pmatrix} \alpha SY + \beta SI \\ \lambda XI \\ 0 \\ 0 \\ 0 \\ 0 \\ 0 \end{pmatrix}, \quad (3)$$

$$\mathscr{V}(\mathscr{X}) = \begin{pmatrix} \theta I + \delta I + \mu_2 I \\ \eta Y + \mu_1 Y \\ -\xi_2 SI - \xi_2 SY - \theta I - \phi D + \mu_2 R \\ -\xi_1 SI - \xi_1 SY - \delta I + \phi D + \mu_2 D \\ H_1 \\ -\Lambda_1 + \lambda SI + \mu_1 X \\ -\eta Y + \mu_1 Z \end{pmatrix} \quad (4)$$

where $H_1 = -\Lambda_2 + \alpha SY + \beta SI + \xi_1 SI + \xi_1 SY + \xi_2 SI + \xi_2 SY + \mu_2 S$.

Therefore

$$F = \begin{pmatrix} \beta \frac{\Lambda_2}{\mu_2} & \alpha \frac{\Lambda_2}{\mu_2} & 0 & 0 \\ \lambda \frac{\Lambda_1}{\mu_1} & 0 & 0 & 0 \\ 0 & 0 & 0 & 0 \\ 0 & 0 & 0 & 0 \end{pmatrix}, \quad (5)$$

$$V = \begin{pmatrix} \theta + \delta + \mu_2 & 0 & 0 & 0 \\ 0 & \eta + \mu_1 & 0 & 0 \\ -\xi_2 \frac{\Lambda_2}{\mu_2} - \theta & -\xi_2 \frac{\Lambda_2}{\mu_2} & \mu_2 & -\phi \\ -\xi_1 \frac{\Lambda_2}{\mu_2} - \delta & -\xi_1 \frac{\Lambda_2}{\mu_2} & 0 & \phi + \mu_2 \end{pmatrix} \quad (6)$$

By calculation we can get

$$FV^{-1} = \begin{pmatrix} \frac{\beta \Lambda_2}{\mu_2(\theta + \delta + \mu_2)} & \frac{\alpha \Lambda_2}{\mu_2(\eta + \mu_1)} & 0 & 0 \\ \frac{\lambda \Lambda_1}{\mu_1(\theta + \delta + \mu_2)} & 0 & 0 & 0 \\ 0 & 0 & 0 & 0 \\ 0 & 0 & 0 & 0 \end{pmatrix} \quad (7)$$

Hence, according to reference [27], the basic reproduction number of system (1) is the spectral radius of matrix $FV^{-1}$ as follows:

$$R_0 = \frac{\beta \Lambda_2}{\mu_2(\theta + \delta + \mu_2)} \quad (8)$$

## B. Existence of equilibrium

According to the system dynamics equation (1), we can calculate the equilibrium $E = (X, Y, Z, S, I, D, R)$. It is easy to see that the positive equilibrium points of system (1) are $E_0 = (\frac{\Lambda_1}{\mu_1}, 0, 0, \frac{\Lambda_2}{\mu_2}, 0, 0, 0)$ and $E^* = (X^*, Y^*, Z^*, S^*, I^*, D^*, R^*)$, and the rumor free equilibrium point $E_0$ always exists.

**Theorem 1** *The equilibrium point $E^* = (X^*, Y^*, Z^*, S^*, I^*, D^*, R^*)$ exists if $R_0 > 1$ and $(\theta + \delta + \mu_2)(\mu_1\beta + \mu_1\xi_1 + \mu_2\lambda) > \beta\lambda\Lambda_2$.*

**Proof** The rumors about system (1) have a balance point that satisfies:

$$\begin{cases} \Lambda_1 - \lambda XI - \mu_1 X = 0, \\ \lambda XI - \eta Y - \mu_1 Y = 0, \\ \eta Y - \mu_1 Z = 0, \\ \Lambda_2 - \alpha SY - \beta SI - (\xi_1 + \xi_2)(I + Y)S - \mu_2 S = 0, \\ \alpha SY + \beta SI - (\theta + \delta)I - \mu_2 I = 0, \\ \xi_1 S(I + Y) + \delta I - \phi D - \mu_2 D = 0, \\ \xi_2 S(I + Y) + \theta I + \phi D - \mu_2 R = 0. \end{cases} \tag{9}$$

According to formula (9), $X^*, Y^*, Z^*, S^*, D^*, R^*$ are represented by $I^*$ respectively and brought into the fifth equation to get

$$aI^2 + bI + c = 0 \tag{10}$$

Where

$$\begin{aligned} a &= \lambda(\beta + \xi_1)(\eta + \mu_1)(\theta + \delta + \mu_2), \\ b &= (\theta + \delta + \mu_2)[(\eta + \mu_1)(\mu_1\beta + \mu_1\xi_1 + \mu_2\lambda)] \\ &\quad + \lambda\Lambda_1(\alpha + \xi_2)(\theta + \delta + \mu_2) - \beta\lambda\Lambda_2(\eta + \mu_1), \\ c &= (\eta + \mu_1)[\mu_2\lambda(\theta + \delta + \mu_2) - \mu_1\beta\Lambda_2] - \alpha\lambda\Lambda_1\Lambda_2. \end{aligned} \tag{11}$$

It can be obtained by calculation that

$$\begin{aligned} \Delta &= b^2 - 4ac \\ &= [\lambda\Lambda_1(\alpha + \xi_2) + (\eta + \mu_1)(\mu_1\beta + \mu_1\xi_1 + \mu_2\lambda)]^2 \\ &\quad * (\theta + \delta + \mu_2)^2 + [\beta\lambda\Lambda_2(\eta + \mu_1)]^2 + 4\alpha\lambda^2\Lambda_1\Lambda_2(\beta + \xi_1) \\ &\quad * (\eta + \mu_1)(\theta + \delta + \mu_2) - 2\beta\Lambda_2(\eta + \mu_1)(\theta + \delta + \mu_2) \\ &\quad * [\lambda\Lambda_1(\alpha + \xi_2) + (\eta + \mu_1)(\mu_1\beta + \mu_1\xi_1 + \mu_2\lambda)] \\ &\quad - 4\lambda(\eta + \mu_1)(\theta + \delta + \mu_2)(\beta + \xi_1)(\eta + \mu_1) \\ &\quad * [\mu_2\lambda(\theta + \delta + \mu_2) - \beta\mu_1\Lambda_2] \end{aligned} \tag{12}$$

According to the discriminant calculation, when $R_0 > 1$ and $(\theta + \delta + \mu_2)(\mu_1\beta + \mu_1\xi_1 + \mu_2\lambda) > \beta\lambda\Lambda_2$, the negative solution is omitted:

$$I^* = \frac{\beta\lambda\Lambda_2(\eta + \mu_1) - H_2(\theta + \delta + \mu_2) + \sqrt{\Delta}}{2\lambda(\beta + \xi_1)(\eta + \mu_1)(\theta + \delta + \mu_2)} \tag{13}$$

where $H_2 = [\lambda\Lambda_1(\alpha + \xi_2) + (\eta + \mu_1)(\mu_1\beta + \mu_1\xi_1 + \mu_2\lambda)]$.

Therefore $E^* = (X^*, Y^*, Z^*, S^*, I^*, D^*, R^*)$, where

$$X^* = \frac{\Lambda_1}{\lambda I^* + \mu_1}, \tag{14}$$

$$Y^* = \frac{\lambda\Lambda_1 I^*}{(\eta + \mu_1)(\lambda I^* + \mu_1)}, \tag{15}$$

$$Z^* = \frac{\lambda\eta\Lambda_1 I^*}{\mu_1(\eta + \mu_1)(\lambda I^* + \mu_1)}, \tag{16}$$

$$S^* = \frac{\Lambda_2(\eta + \mu_1)(\lambda I^* + \mu_1)}{T}, \tag{17}$$

$$D^* = \frac{\lambda\Lambda_2(\eta + \mu_1)I^{*2} + [\lambda\Lambda_1\Lambda_2 + \mu_1(\eta + \mu_1)]I^*}{(\phi + \mu_2)T}, \tag{18}$$

$$R^* = \frac{\xi_2\Lambda_2(\eta + \mu_1)H_3 + \theta H_4}{(\mu_2 - \phi)H_4} \tag{19}$$

where $H_3 = (\lambda I^* + \mu_1)[\lambda\Lambda_1 + (\eta + \mu_1)(\lambda I^* + \mu_1)]$, $H_4 = \lambda(\beta + \xi_1)(\eta + \mu_1)I^{*2} + [\lambda\Lambda_1(\alpha + \xi_2) + (\eta + \mu_1)(\mu_1\beta + \mu_1\xi_1 + \mu_2\lambda)]I^* + \mu_2\lambda(\eta + \mu_1)$.

## C. Stability of equilibrium

**Theorem 2** *The equilibrium point $E_0 = (\frac{\Lambda_1}{\mu_1}, 0, 0, \frac{\Lambda_2}{\mu_2}, 0, 0, 0)$ is locally asymptotically stable if $R_0 < 1$. And the equilibrium point $E_0 = (\frac{\Lambda_1}{\mu_1}, 0, 0, \frac{\Lambda_2}{\mu_2}, 0, 0, 0)$ is unstable if $R_0 > 1$.*

**Proof** The Jacobian matrix of system (1) at $E_0 = (\frac{\Lambda_1}{\mu_1}, 0, 0, \frac{\Lambda_2}{\mu_2}, 0, 0, 0)$ is

$$J(E_0) = \begin{pmatrix} -\mu_1 & 0 & 0 & 0 & -\lambda\frac{\Lambda_1}{\mu_1} & 0 & 0 \\ 0 & -\eta - \mu_1 & 0 & 0 & \lambda\frac{\Lambda_1}{\mu_1} & 0 & 0 \\ 0 & \eta & -\mu_1 & 0 & 0 & 0 & 0 \\ 0 & H_5 & 0 & -\mu_2 & H_6 & 0 & 0 \\ 0 & \alpha\frac{\Lambda_2}{\mu_2} & 0 & 0 & H_7 & 0 & 0 \\ 0 & \xi_2\frac{\Lambda_2}{\mu_2} & 0 & 0 & \xi_2\frac{\Lambda_2}{\mu_2} + \theta & -\mu_2 & -\mu_2 \\ 0 & \xi_1\frac{\Lambda_2}{\mu_2} & 0 & 0 & \xi_1\frac{\Lambda_2}{\mu_2} + \delta & 0 & H_8 \end{pmatrix}$$

where $H_5 = -(\alpha + \xi_1 + \xi_2)\frac{\Lambda_2}{\mu_2}$, $H_6 = -(\beta + \xi_1 + \xi_2)\frac{\Lambda_2}{\mu_2}$, $H_7 = \beta\frac{\Lambda_2}{\mu_2} - (\theta + \delta + \mu_2)$, $H_8 = -(\phi + \mu_2)$.

The characteristic equation of matrix $J(E_0)$ is

$$|J(E_0) - hE| = \begin{vmatrix} T_1 & 0 & 0 & 0 & -\lambda\frac{\Lambda_1}{\mu_1} & 0 & 0 \\ 0 & T_1 & 0 & 0 & \lambda\frac{\Lambda_1}{\mu_1} & 0 & 0 \\ 0 & \eta & T_4 & 0 & 0 & 0 & 0 \\ 0 & T_2 & 0 & T_5 & T_3 & 0 & 0 \\ 0 & \alpha\frac{\Lambda_2}{\mu_2} & 0 & 0 & T_4 & 0 & 0 \\ 0 & \xi_2\frac{\Lambda_2}{\mu_2} & 0 & 0 & T_5 & -\mu_2 - h & -\mu_2 \\ 0 & \xi_1\frac{\Lambda_2}{\mu_2} & 0 & 0 & T_6 & 0 & -(\phi + \mu_2) - h \end{vmatrix}$$
$$= (\mu_1 + h)^2(\mu_2 + h)^2(\phi + \mu_2 + h)(\eta + \mu_1 + h)[\beta\frac{\Lambda_2}{\mu_2} - (\theta + \delta + \mu_2) - h] = 0$$

Where $T_1 = -\mu_1 - h$, $T_2 = -\eta - \mu_1 - h$, $T_3 = -(\alpha + \xi_1 + \xi_2)\frac{\Lambda_2}{\mu_2}$, $T_4 = -\mu_1 - h$, $T_5 = -\mu_2 - h$, $T_6 = -(\beta + \xi_1 + \xi_2)\frac{\Lambda_2}{\mu_2}$, $T_7 = \beta\frac{\Lambda_2}{\mu_2} - (\theta + \delta + \mu_2) - h$, $T_8 = \beta\frac{\Lambda_2}{\mu_2} - (\theta + \delta + \mu_2) - h$, $T_9 = \xi_1\frac{\Lambda_2}{\mu_2} + \delta$.

Therefore, the characteristic root corresponding to the characteristic equation of $J(E_0)$ is:

$$h_{01} = -\mu_1 < 0, h_{02} = -\mu_2 < 0, h_{03} = -(\phi + \mu_2) < 0,$$
$$h_{04} = -(\eta + \mu_1) < 0, h_{05} = \frac{\theta + \delta + \mu_2}{\mu_2}(R_0 - 1) < 0 \tag{20}$$

According to Routh-Hurwitz stability criterion, the equilibrium point
$E_0 = (\frac{\Lambda_1}{\mu_1}, 0, 0, \frac{\Lambda_2}{\mu_2}, 0, 0, 0)$ is locally asymptotically stable if $R_0 < 1$.

And the equilibrium point $E_0 = (\frac{\Lambda_1}{\mu_1}, 0, 0, \frac{\Lambda_2}{\mu_2}, 0, 0, 0)$ is unstable if $R_0 > 1$.

**Theorem 3** *The equilibrium point $E^* = (X^*, Y^*, Z^*, S^*, I^*, D^*, R^*)$ is locally asymptotically stable if $R_0 > 1$ and $\beta\Lambda_2 < \Lambda_1(\alpha + \xi_2)(\theta + \delta + \mu_2)$, otherwise, the equilibrium point $E^*$ is unstable.*

**Proof** The Jacobian matrix at $E^* = (X^*, Y^*, Z^*, S^*, I^*, D^*, R^*)$ is

$J(E^*) =$
$$\begin{pmatrix} A_1 & 0 & 0 & 0 & -\lambda X^* & 0 & 0 \\ \lambda I^* & A_2 & 0 & 0 & \lambda X^* & 0 & 0 \\ 0 & \eta & -\mu_1 & 0 & 0 & 0 & 0 \\ 0 & A_3 & 0 & A_4 & A_8 & 0 & 0 \\ 0 & \alpha S^* & 0 & A_5 & A_9 & 0 & 0 \\ 0 & \xi_2 S^* & 0 & A_6 & \xi_2 S^* + \theta & -\mu_2 & -\mu_2 \\ 0 & \xi_1 S^* & 0 & A_7 & \xi_1 S^* + \delta & 0 & A_{10} \end{pmatrix}$$

Where $A_1 = \lambda I^* - \mu_1$, $A_2 = -\eta - \mu_1$, $A_3 = -(\alpha + \xi_1 + \xi_2)S^*$, $A_4 = -\alpha Y^* - \beta I^*$, $A_5 = \alpha Y^* + \beta I^*$, $A_6 = \xi_2(I^* + Y^*)$, $A_7 = \xi_1(I^* + Y^*)$, $A_8 = -(\beta + \xi_1 + \xi_2)S^*$, $A_9 = \beta S^* - (\theta + \delta + \mu_2)$, $A_{10} = -(\phi + \mu_2)$.

The characteristic equation of matrix $J(E^*)$ is
$|J(E^*) - hE| =$
$$\begin{vmatrix} B_1 & 0 & 0 & 0 & -\lambda X^* & 0 & 0 \\ \lambda I^* & B_2 & 0 & 0 & \lambda X^* & 0 & 0 \\ 0 & \eta & -\mu_1 - h & 0 & 0 & 0 & 0 \\ 0 & B_3 & 0 & B_3 & B_7 & 0 & 0 \\ 0 & \alpha S^* & 0 & B_4 & B_8 & 0 & 0 \\ 0 & \xi_2 S^* & 0 & B_5 & \xi_2 S^* + \theta & B_9 & -\mu_2 \\ 0 & \xi_1 S^* & 0 & B_6 & \xi_1 S^* + \delta & 0 & B_{10} \end{vmatrix}$$

Where $B_1 = \lambda I^* - \mu_1 - h$, $B_2 = -\eta - \mu_1 - h$, $B_3 = -\alpha Y^* - \beta I^* - h$, $B_4 = \alpha Y^* + \beta I^*$, $B_5 = \xi_2(I^* + Y^*)$, $B_6 = \xi_1(I^* + Y^*)$, $B_7 = -(\beta + \xi_1 + \xi_2)S^*$, $B_8 = \beta S^* - (\theta + \delta + \mu_2) - h$, $B_9 = -\mu_2 - h$, $B_{10} = -(\phi + \mu_2) - h$.

Thus, we can obtain
$|J(E^*) - hE| = (\mu_1 + h)(\mu_2 + h)(\phi + \mu_2 + h)(\eta + \mu_1 + h)(\lambda I^* + \mu_1 + h)G$.

Where $G = -[\alpha Y^* + \beta I^* + (\xi_1 + \xi_2)(I^* + Y^*) + \mu_2] - h$.

Therefore, the characteristic root corresponding to the characteristic equation of $J(E^*)$ is:

$$h_{01} = -\mu_1 < 0, h_{02} = -\mu_2 < 0, \tag{21}$$

$$h_{03} = -(\phi + \mu_2) < 0, h_{04} = -(\eta + \mu_1) < 0, \tag{22}$$

$$h_{05} = -[\alpha Y^* + \beta I^* + (\xi_1 + \xi_2)(I^* + Y^*) + \mu_2] < 0, \tag{23}$$

$$h_{06} = \beta S^* - (\theta + \delta + \mu_2). \tag{24}$$

Then, we take $S^*$ into $h_{06}$,
$h_{06} = \frac{\beta\Lambda_2(\eta + \mu_1)(\lambda I^* + \mu_1)}{\lambda(\beta + \xi_1)(\eta + \mu_1)I^{*2} + C_1 + \mu_2\lambda(\eta + \mu_1)} - (\theta + \delta + \mu_2)$,
*where* $C_1 = [\lambda\Lambda_1(\alpha + \xi_2) + (\eta + \mu_1)(\mu_1\beta + \mu_1\xi_1 + \mu_2\lambda)]I^*$.

## IV. NUMERICAL SIMULATION

In this section, we will assign reasonable values to the parameters in system (1) as established in this paper, and verify the results of our theoretical analysis through numerical simulations. On the one hand, we combine some similar examples in reality. On the other hand, the parameter values in relevant literature are referred to.

Order $\Lambda_1 = 1, \Lambda_2 = 1, \lambda = 0.01, \eta = 0.3, \alpha = 0.01, \beta = 0.01, \theta = 0.2, \delta = 0.2, \phi = 0.15, \xi_1 = 0.1, \xi_2 = 0.1, \mu_1 = 0.2, \mu_2 = 0.2$. Calculated $R_0 = 0.8333 < 1$, then the no rumor propagation equilibrium point $E_0$ is stable.

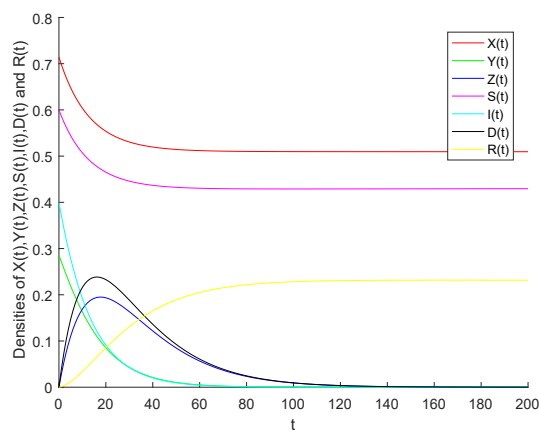

Fig 2. Stability of equilibrium point $E_0$.

Fig. 2 shows when $R_0 = 0.0833 < 1$, the density of each subclass in the model changes with time. At first, the number of unaffected media and unknowns gradually decreased at a similar rate, and finally stabilized. Due to the limited number of moving in and the large number of moving out, the number of affected media and communicators gradually decreases at a similar rate and finally becomes 0. The number of rumor refuting media and rumor refuters first increased with the increase of the number of affected media and disseminators. It gradually decreases over time and finally becomes 0. The number of immunized persons increased with the increase of the number of communicators and rumor refuters. The growth rate gradually slowed down and finally stabilized. Namely, the rumor disappears and reaches a stable equilibrium point, and there is no rumor.

Let $\Lambda_1 = 1, \Lambda_2 = 1, \lambda = 0.2, \eta = 0.3, \alpha = 0.5, \beta = 0.6, \theta = 0.4, \delta = 0.4, \phi = 0.15, \xi_1 = 0.2, \xi_2 = 0.2, \mu_1 =$

$0.2, \mu_2 = 0.2$ and calculate $R_0 = 3 > 1$, the equilibrium point $E^*$ is stable, as shown in Fig. 3.

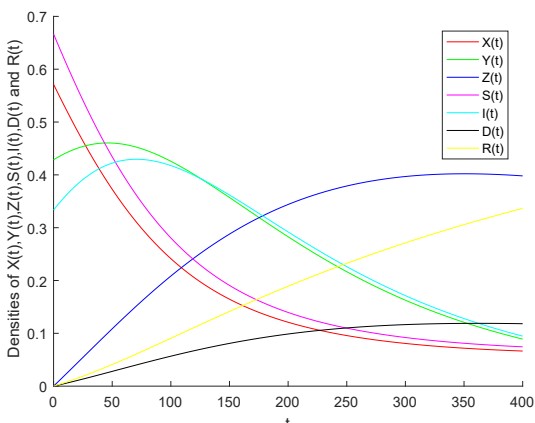

Fig 3. Stability of equilibrium point $E^*$.

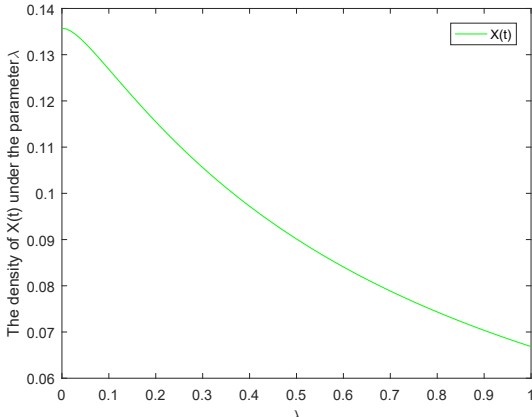

Fig 4. Density of $X(t)$ under the parameter $\lambda$.

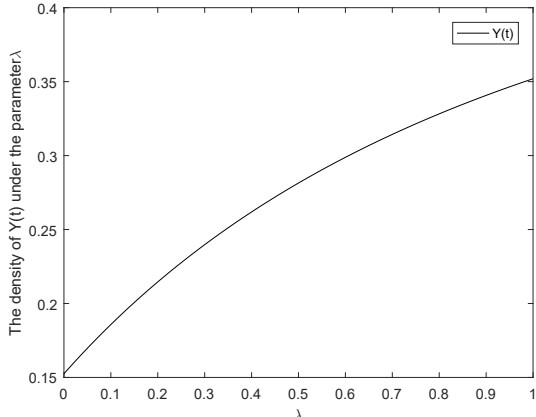

Fig 5. Density of $Y(t)$ under the parameter $\lambda$.

In Fig. 3, considering the media network layer, due to the small number of new media entering the communication system and the transformation of some unaffected media into affected media, the number of unaffected media gradually decreases and tends to stabilize after a period of time. Originally, the number of affected media increased due to the transformation of some unaffected media into affected media. Over time, most of the affected media were transformed into rumor refuting media, so the number of affected media decreased and stabilized. As the affected media changed into rumor refuting media, the number of rumor refuting media increased and gradually stabilized.

Fig. 3 illustrates that, within the individual interpersonal network layer, the number of ignorant individuals begins to decline. Initially, the low influx of new individuals and a fixed rate of departures contribute to this decrease. Additionally, some ignorant individuals transition to become communicators, while others become immune or rumor refuters. Consequently, the number of communicators increases as ignorant individuals transform into communicators. Over time, as communicators transition to immune individuals or rumor refuters, the number of communicators gradually decreases and eventually stabilizes. As more communicators and ignorant individuals become rumor refuters, the number of rumor refuters rises and stabilizes. Simultaneously, with some ignorant individuals, communicators, and rumor refuters becoming immune, the number of immune individuals significantly increases and gradually stabilizes. Ultimately, the model reaches a steady state, with each groups number stabilizing over time.

Fig. 4 to Fig. 7 depict $\Lambda_1 = 1, \Lambda_2 = 1, \eta = 0.3, \theta = 0.4, \phi = 0.15, \xi_1 = 0.2, \xi_2 = 0.5, \mu_1 = 0.2, \mu_2 = 0.2$, the evolution of the density of $X(t), Y(t)$ and $S(t)$ with different parameters.

Fig. 4 and Fig. 5 describe the effect of parameter $\lambda$ on the density change of $X(t)$ and $Y(t)$ respectively. Parameter $\lambda$ represents the probability that the unaffected media will be transformed into the affected media. It can be seen from the figure that the parameter $\lambda$ has a negative correlation with the density of $X(t)$ and a positive correlation with the density of $Y(t)$. That is, with the increase of the parameter $\lambda$, the rate of transformation from unaffected media to affected media increases. The number of unaffected media decreases gradually, and the number of affected media increases gradually, accelerating the spread of rumors in the media network layer.

Fig. 6 and Fig. 7 describe the influence of parameters $\alpha$ and $\beta$ on the density change of $S(t)$ respectively. Parameter $\alpha$ represents the probability that the unknown person will become a spreader by accessing the affected media, and parameter $\beta$ represents the probability that the unknown person will become a spreader by contacting spreaders. It can be seen from the figure that the density of $S(t)$ decreases with the increase of parameters $\alpha$ and $\beta$. Namely, with the increase of the propagation rate of individual network layer and double-layer network interaction, the number of unknowns gradually decreases, which accelerates the spread of rumors in the double-layer network.

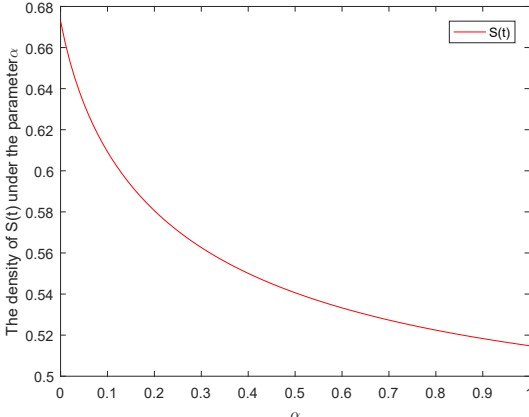

Fig 6. Density of $S(t)$ under the parameter $\alpha$.

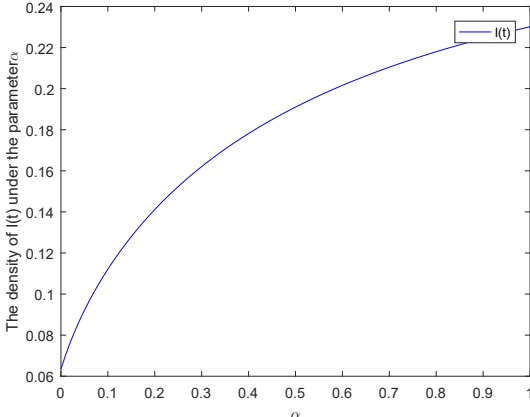

Fig 8. Density of $I(t)$ under the parameter $\alpha$.

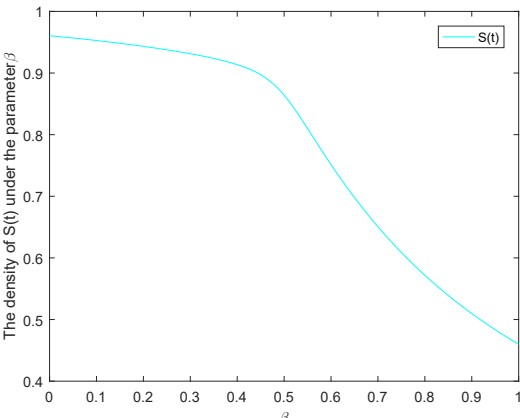

Fig 7. Density of $S(t)$ under the parameter $\beta$.

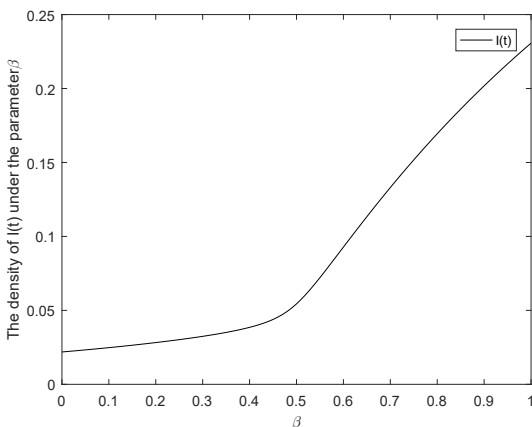

Fig 9. Density of $I(t)$ under the parameter $\beta$.

Fig. 8 and Fig. 9 describe when $\Lambda_1 = 1, \Lambda_2 = 1, \eta = 0.3, \theta = 0.4, \phi = 0.15, \xi_1 = 0.2, \xi_2 = 0.5, \mu_1 = 0.2, \mu_2 = 0.2$, the evolution of the density of $I(t)$ with different parameters.

Fig. 8 and Fig. 9 describe the influence of parameters $\alpha$ and $\beta$ on the density change of $I(t)$ respectively. Considering the meaning of parameters $\alpha$ and $\beta$, it is easy to know that the values of parameters $\alpha$ and $\beta$ are positively correlated with the density of $I(t)$. As can be seen from the figure, the density of $I(t)$ increases with the increase of parameters $\alpha$ and $\beta$. That is, the increasing number of communicators promotes the expansion of the scale of communication, which is not conducive to the control of rumors.

## V. CONCLUSION

At present, many scholars have separately studied the influence of media refutation or individual refutation on the spread of rumors. We believe that considering these two effects comprehensively is better than considering one of them alone. This paper integrates both media refutation and individual refutation into the analysis and introduces a novel $XYZ - SIDR$ two-tier rumor propagation model, further demonstrates the existence and stability of equilibrium points within the model. The research results show that this two-layer network model is more effective in controlling the spread of rumors.

Theoretical analysis indicates that integrating both media and individual rumor refutation exerts a more significant and broader impact on rumor propagation. We suggest strengthening the dissemination of rumor refutation information through the official media rather than relying solely on individuals to control the spread of rumor. The research conclusion can help relevant departments formulate effective measures to control the spread of rumors. On the other hand, the model established in this paper can also be analogically applied to the study of infectious disease model.

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
