# OpenReview forum: "Dynamical analysis of rumor propagation model considering media refutation and individual refutation"
_IEEE.org/ICIST/2024/Conference — IEEE ICIST 2024 Conference Submission_

### Official Review · Reviewer_aNiZ · 2024-08-21
**This paper is novel and of practical value, and is recommended for publication.**

**Rating:** 7
**Confidence:** 4

**Review:**

This paper provides a compelling overview of a study that seeks to address a gap in existing research on rumor refutation. By proposing a two-tier network model, the study aims to integrate both media and individual refutations in understanding rumor propagation, a novel approach that promises to offer deeper insights into the dynamics of rumor control. The finding that authoritative media refutations have a broader and more significant impact than individual refutations adds a valuable dimension to the discussion on effective rumor management.

1.Can you provide more details about the two-tier network model? How are media and individual refutations represented within this model, and what specific mechanisms are used to simulate their effects?

2.What are the characteristics of the equilibrium points identified in the model? How do these points influence the overall stability and effectiveness of rumor refutation strategies?

3.How did the study measure and compare the impacts of media versus individual refutations? Are there specific metrics or data that demonstrate the broader influence of media interventions?

4.What are the practical implications of your findings for real-world rumor management? How can organizations or policymakers apply this model to improve their strategies for controlling rumor spread?

5.Are there any limitations or assumptions in your model that might affect its applicability to different types of rumors or contexts? What further research is needed to build on your findings?

---

### Official Review · Reviewer_PPTK · 2024-08-21
**Accept**

**Rating:** 7
**Confidence:** 4

**Review:**

This paper discusses a dynamic analysis of rumor propagation, considering the effects of both media refutation and individual refutation. It proposes a novel two-tier network model for rumor spread, integrating the influence of media and individual refutation. The study demonstrates the existence and stability of equilibrium points within the model and suggests that a comprehensive approach to rumor refutation, involving both media and individuals, is more effective than considering one factor alone.
﻿
1.How does the interaction between media refutation and individual refutation affect the overall spread of rumors in the two-tier network model?

2.What are the key parameters that influence the stability of the equilibrium points in the rumor propagation model?

3.In the proposed model, how do changes in the rate of individual refutation impact the density of rumor spreaders over time?

4.How can the findings from this study be applied to real-world scenarios to help control the spread of rumors more effectively?

---

### Official Review · Reviewer_9WNf · 2024-08-22
**Interesting study**

**Rating:** 7
**Confidence:** 4

**Review:**

The paper presents a significant contribution to the field of rumor propagation modeling by introducing a novel two-tier network model that considers both media and individual refutation. While the research is grounded in a strong theoretical framework and supported by numerical simulations, the abstract could benefit from improved clarity, a more detailed comparison with existing methods, and a brief discussion of limitations. Addressing these points would enhance the overall impact and readability of the paper.

---

### Decision · Program_Chairs · 2024-09-08

Accept (Oral)